# Ecdysone acts through cortex glia to regulate sleep in *Drosophila*

Yongjun Li[1,2], Paula Haynes[1,3], Shirley L Zhang[1†], Zhifeng Yue[1], Amita Sehgal[1]*

[1]Howard Hughes Medical Institute and Chronobiology and Sleep Institute, Perelman School of Medicine at the University of Pennsylvania, Philadelphia, United States; [2]Department of Biology, University of Pennsylvania, Philadelphia, United States; [3]Department of Pharmacology, Perelman School of Medicine at the University of Pennsylvania, Philadelphia, United States

**Abstract** Steroid hormones are attractive candidates for transmitting long-range signals to affect behavior. These lipid-soluble molecules derived from dietary cholesterol easily penetrate the brain and act through nuclear hormone receptors (NHRs) that function as transcription factors. To determine the extent to which NHRs affect sleep:wake cycles, we knocked down each of the 18 highly conserved NHRs found in *Drosophila* adults and report that the ecdysone receptor (EcR) and its direct downstream NHR Eip75B (E75) act in glia to regulate the rhythm and amount of sleep. Given that ecdysone synthesis genes have little to no expression in the fly brain, ecdysone appears to act as a long-distance signal and our data suggest that it enters the brain more at night. Anti-EcR staining localizes to the cortex glia in the brain and functional screening of glial subtypes revealed that EcR functions in adult cortex glia to affect sleep. Cortex glia are implicated in lipid metabolism, which appears to be relevant for actions of ecdysone as ecdysone treatment mobilizes lipid droplets (LDs), and knockdown of glial EcR results in more LDs. In addition, sleep-promoting effects of exogenous ecdysone are diminished in *lsd-2* mutant flies, which are lean and deficient in lipid accumulation. We propose that ecdysone is a systemic secreted factor that modulates sleep by stimulating lipid metabolism in cortex glia.

## Editor's evaluation

This is a strong manuscript that identified a role for ecdysone signaling and cortex glia in sleep regulation. The manuscript is important because it opens up new avenues of study for examining how hormone signaling and glia regulate sleep and circadian rhythms.

## Introduction

Sleep is a resting state that alternates with awake and is broadly conserved in animals (**Anafi et al., 2018**). With recent advances in our understanding of sleep regulation and function, much of it from small animal models, it is becoming increasing apparent that sleep is more than a brain-regulated process that only serves the brain (**Borbély et al., 2016**; **Dubowy and Sehgal, 2017**; **Harbison et al., 2009**; **Hobson, 2005**; **Shaw et al., 2000**). Loss of sleep has systemic effects and is associated with molecular changes in the periphery (**Anafi et al., 2013**; **Chua et al., 2015**). In addition, some sleep-promoting effects have been mapped to tissues outside the brain, although the nature of the sleep-regulating signals and their mode of transmission to relevant cells in the brain are poorly understood (**Borniger and de Lecea, 2021**; **Ehlen et al., 2017**; **Toda et al., 2019**).

As a major communication system of the body, endocrine signaling is a promising candidate for mediating long-range effects on sleep (**Mong et al., 2011**; **Morgan and Tsai, 2015**; **Roller, 2021**).

*For correspondence: amita@pennmedicine.upenn.edu

Present address: †Department of Cell Biology, Emory University School of Medicine, Atlanta, United States

**Competing interest:** The authors declare that no competing interests exist.

Glands of the endocrine system make and release chemical messengers called steroid hormones, which circulate in the blood until they reach their target cells and bind to specific steroid nuclear hormone receptors (NHRs) in the cytosol (*Bonora and DeFronzo, 2018*; *Zubeldia-Brenner et al., 2016*). Given that circulating steroid hormones are lipophilic and can easily enter the brain, together with the broad expression of NHRs in the central nervous system, it is reasonable to infer that NHR signaling is a significant contributor to brain function (*Yang et al., 2007*; *King-Jones and Thummel, 2005*; *Kininis and Kraus, 2008*). However, the response mediated by the endocrine system is modulatory, in that it is slower and more long term than fast, synaptic neurotransmission (*Morgan and Tsai, 2015*). Thus, the endocrine system drives responses to environmental cues and innate signals critical for development, growth, and metabolism (*Kannangara et al., 2021*; *Oostra et al., 2014*). Sleep is a chronic behavior regulated over a relatively long timeframe (a daily cycle), so one might expect it also to be sensitive to endocrine signaling. Indeed, endocrine dysfunction affects sleep, and conversely, sleep and sleep loss affect hormone production and hormonal function, but the specificity and mechanisms underlying these interactions remain to be elucidated (*Aldabal and Bahammam, 2011*; *Kim et al., 2015*; *Terán-Pérez et al., 2012*).

To explore the impact of endocrine signaling on sleep, we used a *Drosophila* model and knocked down each of the known *Drosophila* NHRs in neurons and glia. We found that the ecdysone receptor (EcR) and Eip75B (E75) function in both cell types to modulate sleep but have more potent effects in glia. EcR knockdown in glia significantly decreases sleep and disrupts circadian rhythms, as does the knockdown of the newly identified ecdysone importer (EcI). EcR functions in cortex glia to affect sleep, and it appears to do so by regulating lipid metabolism. Together these studies identify a long-range signal that modulates sleep in adult *Drosophila*.

## Results
### Effects of pan-neuronal and pan-glial knockdown of EcR and E75 on sleep

To identify NHRs that affect sleep in adult *Drosophila*, we conducted a systematic genetic knockdown screen of all known NHRs. To avoid developmental effects, which are prominent for most NHRs, we sought to address their role in sleep by knocking them down specifically at the adult stage (*Nicholson et al., 2008*). Based upon single-cell RNAseq data showing that nearly half of known NHRs are highly expressed in adult fly brain both neurons and glia, we used pan-neuronal and pan-glial GeneSwitch (GS) drivers induced in adults with RU486 (nSyb-GS and Repo-GS, respectively) to knock down individual NHRs (*Davie et al., 2018*; *Nicholson et al., 2008*). All NHRs were targeted either by two separate RNAi lines or UAS-miRNA transgenic fly lines that polycistronically express two independent miRNAs in the same line (*Lin et al., 2009*). Surprisingly, we found that knockdown of most NHRs in neurons and glial cells reduced adult sleep by varying levels compared with their GS controls and UAS controls, suggesting that the endocrine system in general plays a positive role in sleep regulation (*Figure 1B,C*). Among them, knockdown of EcR, the receptor of the primary steroid hormone ecdysone, and its direct downstream targeting gene Eip75B (E75) had the greatest effects on total sleep. Because we noticed that Repo-GS control flies have variable daytime sleep and RU486 induction further reduces daytime sleep, we also analyzed all flies' nighttime sleep, which is quite stable in all conditions. Importantly, EcR and E75 knockdown affect primarily nighttime sleep and phenotypes are stronger when they are knocked down in glia.

Across multiple experiments, we found that flies with glial knockdown of EcR slept on average ~460 min, much less than those with neuronal knockdown that slept ~707 min. In addition, the total sleep amount of nSyb-GS>EcR RNAi flies was not significantly reduced compared with nSyb-GS control flies, suggesting that glial EcR has a larger role in determining sleep amount (*Figure 2A–D*). Knockdown of E75 in either neurons or glia led to severe sleep loss, with ~419 and ~461 min of daily sleep, respectively (*Figure 2—figure supplement 1A–D*). Because ecdysone is the most common steroid hormone in *Drosophila*, it directly or indirectly affects multiple NHRs (*King-Jones and Thummel, 2005*). Indeed, we found that knockdown of ecdysone responsive NHRs reduced sleep more than knockdown of non-ecdysone responsive NHRs, and from the screen of neuronal and glial knockdown, top hits were known ecdysone relevant NHRs (*Figure 1—source data 1*). If we use 200-min sleep loss compared with both control groups as a cutoff, then two out of three top hits with neuronal

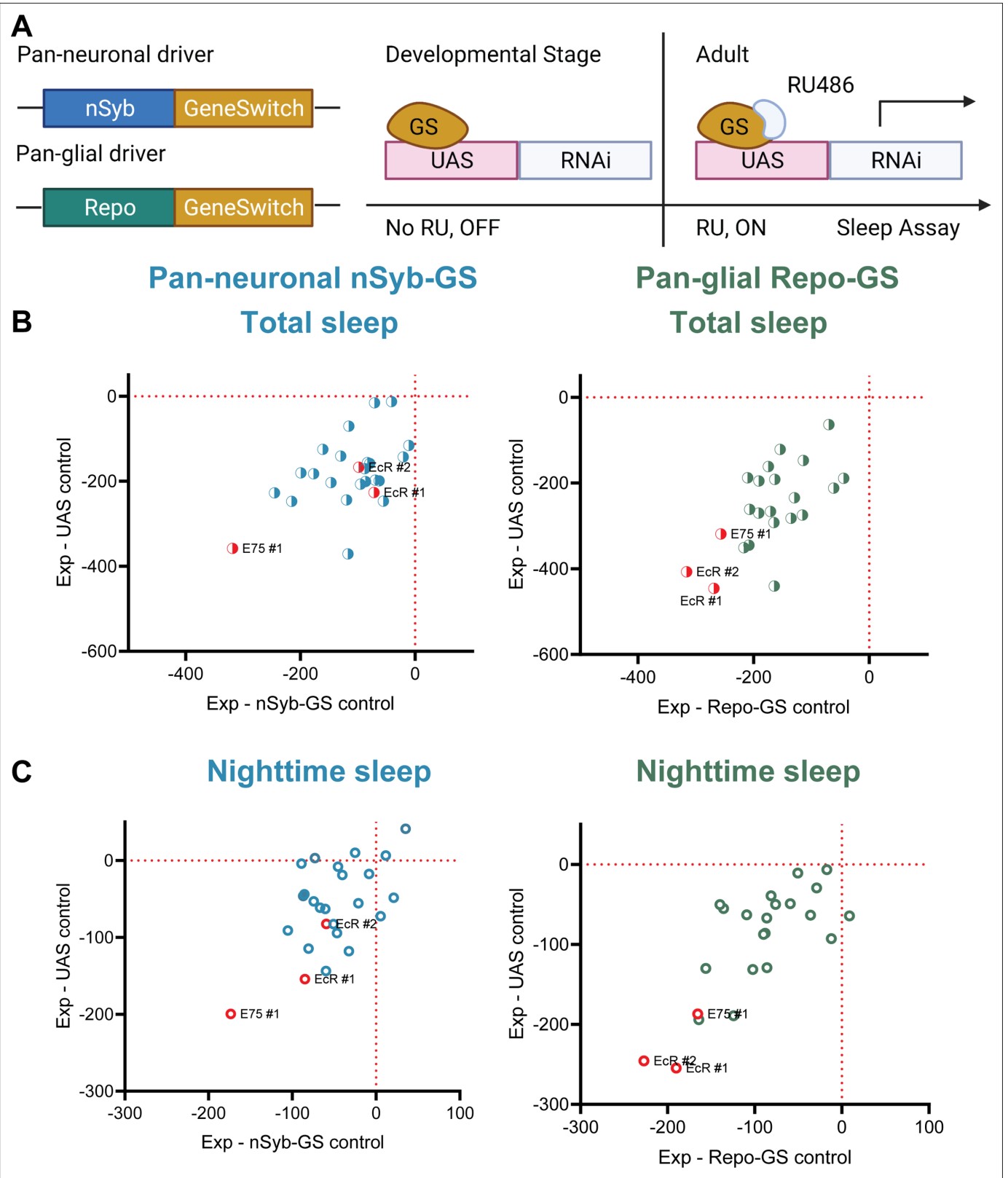

**Figure 1.** A screen of all nuclear hormone receptors (NHRs) in *Drosophila* identifies sleep-regulating functions of ecdysone receptor (EcR) and its downstream target, NHR E75. (**A**) Flies carrying a pan-neuronal driver nSyb-GS or pan-glial driver Repo-GS were crossed with different UAS lines carrying RNAi constructs against NHRs, and their 5- to 7-day-old F1 female progeny were loaded into DAM monitors to record their activity under 12:12 hr light:dark cycles. GeneSwitch remains inactive during developmental stages and is activated by RU486 in the food in DAM monitors. Behavior

*Figure 1 continued on next page*

Figure 1 continued

data were collected by the DAM system. (**B, C**) Mean total and nighttime sleep for each group were calculated by Pysolo. Differences between experimental flies and GS and RNAi controls were calculated separately in each independent experiment, and the average values comparing each experimental to its GeneSwitch control (*X*-axis) and RNAi control (*Y*-axis) are shown in the plots. The exact numbers of flies used per line are provided in *Figure 1—source data 1*. While knockdown of most NHRs reduces sleep, effects of EcR RNAi #1, EcR RNAi #2, E75 RNAi #1 predominate, especially in glial knockdown experiments.

The online version of this article includes the following source data for figure 1:

**Source data 1.** Detailed sleep phenotypes of nuclear hormone receptor (NHR) RNAi screening lines.

knockdown and five out of seven top hits with glial knockdown are ecdysone relevant NHRs. Genes showing the most dramatic sleep loss when knocked down in glia were EcR, E75, Ftz-f1, and Hr3, all directly relevant to the ecdysone signaling pathway. The efficacy of our screen was supported by the identification of non-ecdysone responsive NHR Hr51 (*unfulfilled*) as a hit affecting circadian rhythms. We found that neuronal, but not glial, knockdown of Hr51 (*unfulfilled*), previously shown to exert circadian effects by acting in central clock neurons (*Beuchle et al., 2012*), resulted in an arrhythmic phenotype, with no sleep amount change under 12:12 hr light:dark conditions (*Figure 2—figure supplement 2A–D*).

Sleep architecture analysis showed that neuronal or glial-specific knockdown of EcR or E75 reduces sleep amount by fragmenting sleep, with larger effects seen with the glial-specific knockdown (*Figure 2E–H*, *Figure 2—figure supplement 1E–H*). Glial EcR and E75 knockdown flies had shorter average sleep bout lengths during the day and night and more sleep bouts at night, indicating that their sleep is fragmented (*Figure 2F, H* and *Figure 2—figure supplement 1F, H*). Comparison of RU486-treated flies with vehicle-treated controls confirmed that adult-specific knockdown of EcR is sufficient to reduce sleep in both neuronal and glial populations (*Figure 2—figure supplement 2E–H*). To confirm effects of adult-specific knockdown, we used the temporal and regional gene expression targeting (TARGET) system in which expression of Repo-Gal4 can be restricted to adults by using a temperature-sensitive allele of a Gal4 suppressor, Gal80 (*McGuire et al., 2003*). The results show that Repo-Gal4; TubGal80ts>EcR RNAi #1 flies sleep a similar amount at permissive temperature (18 degrees) as control flies, but their sleep is significantly less at restricted temperature (31 degrees), confirming that EcR acts in adult glia to regulate daily sleep (*Figure 2—figure supplement 3A,B*).

Since EcR RNAi #1 and #2 used in the initial screening are two independent P-element transgenic lines but were generated using the same construct, we also tested three other EcR RNAi lines with Repo-GS, all of which produced reduced sleep phenotypes, further confirming that EcR RNAi knockdown in glial cells reduces sleep (*Figure 2—figure supplement 3C–E*). Furthermore, we tested the function of the recently identified ecdysone membrane transporter EcI, whose expression is necessary for ecdysone uptake by cells (*Okamoto et al., 2018*). As with EcR, glial knockdown of EcI reduced overall sleep levels and led to more fragmented sleep (*Figure 2—figure supplement 4A–D*). In general, all these data strongly indicate that ecdysone signaling acts in the brain, particularly in glia, to modulate sleep.

To assay flies for free-running circadian rhythms, we monitored them under constant darkness (DD) conditions. In DD, half of the nSyb-GS EcR knockdown flies and 90% of Repo-GS EcR knockdown flies became arrhythmic, but the remaining rhythmic flies showed no difference from controls in their period or FFT (Fast Fourier Transform) values that are a measure of rhythm strength (*Figure 2I–K*). Our finding that EcR functions are required in glia to promote circadian locomotor rhythms and sleep is novel and consistent with previous work showing that E75 or EcR knockdown in central clock cells causes some locomotor arrhythmicity, but more dramatic phenotypes result from knockdown with drivers such as *tim²⁷*-Gal4 that additionally express in glia (*Kumar et al., 2014*).

Next, we overexpressed EcR and found that, in contrast to our knockdown experiments, overexpression in neurons and glia had no effect on total sleep, but Repo-GS>EcR_c flies showed a consistent phase shift at dusk (*Figure 2—figure supplement 4E–H*). Thus, effects of EcR on sleep and rhythms are largely through glia, and not neurons. Regarding the lack of sleep phenotypes from overexpression, it is possible that sleep-relevant ecdysone signaling is saturating in both neurons and glia and increasing EcR does not confer new sleep-promoting functions. However, since exogenous ecdysone feeding promotes sleep (*Ishimoto and Kitamoto, 2010*), it is also possible that the EcR ligand ecdysone is rate limiting when EcR is overexpressed.

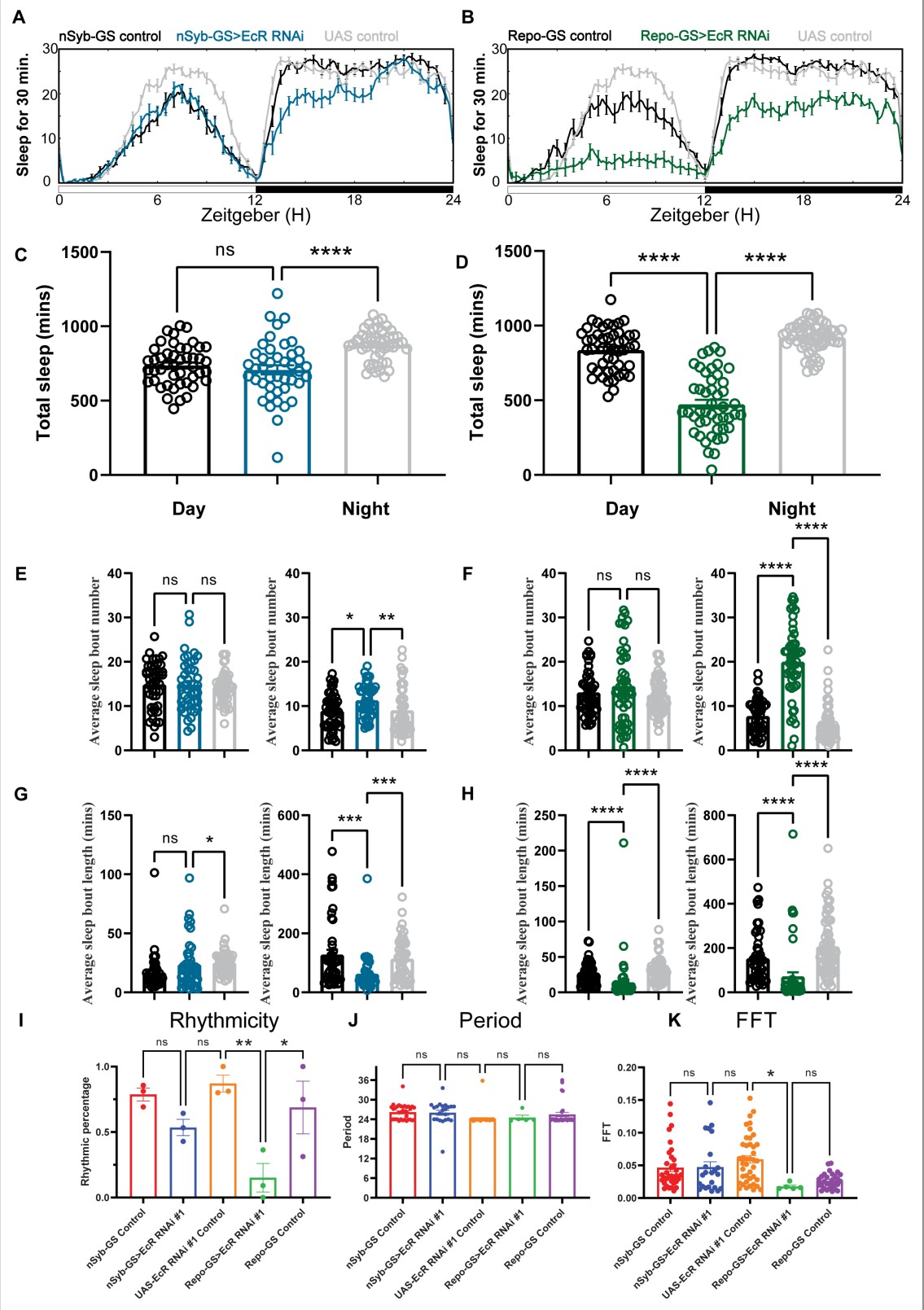

**Figure 2.** Baseline sleep phenotypes resulting from pan-neuronal or pan-glial knockdown of ecdysone receptor (EcR). (**A, B**) Show representative sleep traces of nSyb-GS>EcR RNAi #1 and Repo-GS>EcR RNAi #1. *N* = 14–16 per genotype. Data are based on at least three independent experiments. Representative sleep traces are showed because Pysolo we used does not allow combining data from repeated experiments. Total sleep of the nSyb-GS>EcR RNAi #1 and Repo-GS>EcR RNAi #1 flies for three replicates, *N* = 43–63, one-way analysis of variance (ANOVA) with Tukey post hoc test

*Figure 2 continued on next page*

*Figure 2 continued*

was used for (**C**), and Kruskal–Wallis test with Dunn's multiple comparisons test was used for (**D**). (**E–H**) The average sleep bout number and average sleep bout length of the nSyb-GS>EcR RNAi #1 and Repo-GS>EcR RNAi #1 flies for all three replicates. Daytime sleep data are quantified in the left panels, and nighttime sleep is quantified in the right panels of each group. One-way ANOVA analysis with Tukey post hoc test was used for (**E, F**) and Kruskal–Wallis test with Dunn's multiple comparisons test was used for (**G, H**). (**I–K**) The rhythmicity, period, and relative Fast Fourier Transform (FFT) power analysis of nSyb-GS>EcR RNAi #1 and Repo-GS>EcR RNAi #1 flies and controls assayed for locomotor activity rhythms in constant darkness. (**I**) shows the percentage rhythmic in each genotype from all three independent replicates. Flies used for analysis in (**J, K**) are rhythmic flies from (**I**), and experimental files are compared with their GeneSwitch and RNAi control flies. Bar graphs show mean + standard error of the mean (SEM), and p values for each comparison were calculated using the Kruskal–Wallis test with Dunn's multiple comparisons test. ns = not significant, p > 0.05, *p < 0.05, **p < 0.01, ***p < 0.001, ****p < 0.0001. See also *Figure 2—source data 1*.

The online version of this article includes the following source data and figure supplement(s) for figure 2:

**Source data 1.** Sleep and circadian phenotypes resulting from pan-neuronal or pan-glial knockdown of ecdysone receptor (EcR).

**Source data 2.** Sleep phenotypes resulting from pan-neuronal or pan-glial knockdown of E75.

**Source data 3.** Sleep phenotypes resulting from pan-neuronal or pan-glial knockdown of Hr51 and sleep phenotypes of adult-specific neuronal and glial knockdown of ecdysone receptor (EcR).

**Source data 4.** Sleep phenotypes resulting from pan-glial knockdown of ecdysone receptor (EcR) by temporal and regional gene expression targeting (TARGET) system and different EcR RNAi lines.

**Source data 5.** Sleep phenotypes resulting from pan-neuronal or pan-glial knockdown of EcI and overexpression of ecdysone receptor (EcR) common isoforms.

**Figure supplement 1.** Baseline sleep phenotypes resulting from pan-neuronal and pan-glial knockdown of E75, the direct downstream target of ecdysone receptor (EcR).

**Figure supplement 2.** Adult-specific neuronal and glial knockdown of ecdysone receptor (EcR) reduces sleep.

**Figure supplement 3.** Adult-specific knockdown of ecdysone receptor (EcR) by using the temporal and regional gene expression targeting (TARGET) system reduces sleep.

**Figure supplement 4.** Baseline sleep phenotypes resulting from pan-neuronal and pan-glial overexpression of ecdysone receptor (EcR) common isoforms and knockdown of EcI, the membrane importer of ecdysone.

## Starvation and glial ecdysone signaling act independently to regulate sleep amount

Sleep changes under stress conditions, and ecdysone is implicated in stress responses (*Hill et al., 2018*; *Ishimoto and Kitamoto, 2011*; *Williams, 2019*), raising the possibility that ecdysone mediates the effects of stressors on sleep. Starvation is an example of a stressor that raises ecdysone levels in female flies (*Terashima et al., 2005*). Although its effect on sleep goes in the opposite direction, that is, starvation reduces sleep, we asked whether ecdysone is relevant for effects of starvation (*Keene et al., 2010*) for instance, does downregulation of ecdysone signaling in specific cells (despite overall increased ecdysone) account for decreased sleep during starvation? As shown by *Ishimoto and Kitamoto, 2010*, we also found that feeding of the exogenous bioactive form of ecdysone, 20E, promotes sleep in fed wild-type flies (*Figure 3 A, E*); we then asked if starved flies respond to 20E as well. Exogenous ecdysone abrogated sleep loss caused by starvation, indicating that EcRs are functional in sleep-relevant cells under these conditions (*Figure 3B, E*). Consistent with this, we found that mRNA levels of ecdysone responsive genes—EcR, EcI, E75, and E74—in the fly brain are not altered after 1 day of starvation (*Figure 3—figure supplement 2D*). Interestingly, starved flies fed 20E showed rebound sleep when they were returned to food, even though these flies did not lose sleep during starvation, indicating that 20E does not rescue the need for sleep that builds up during starvation (*Regalado et al., 2017*).

To determine whether increased sleep after 20E feeding is mediated by neurons or glia, we knocked down EcR in each of those two cell types. Sleep increased following 20E administration in nSyb-GS>EcR RNAi flies, but not Repo-GS>EcR RNAi flies (*Figure 3—figure supplement 1A*). As the Repo-GS control flies also did not respond to 20E feeding, we cannot draw conclusions about the necessity of EcR expression in glia for effects of exogenous ecdysone. However, neuronal ecdysone signaling appears to be dispensable for this purpose.

We next asked how knockdown of EcR in neurons versus glia effects the response to starvation and found that while starvation reduced sleep in flies in which EcR was knocked down in neurons, the effects was not significant (*Figure 3C, E*). Although this could suggest a role of neuronal ecdysone

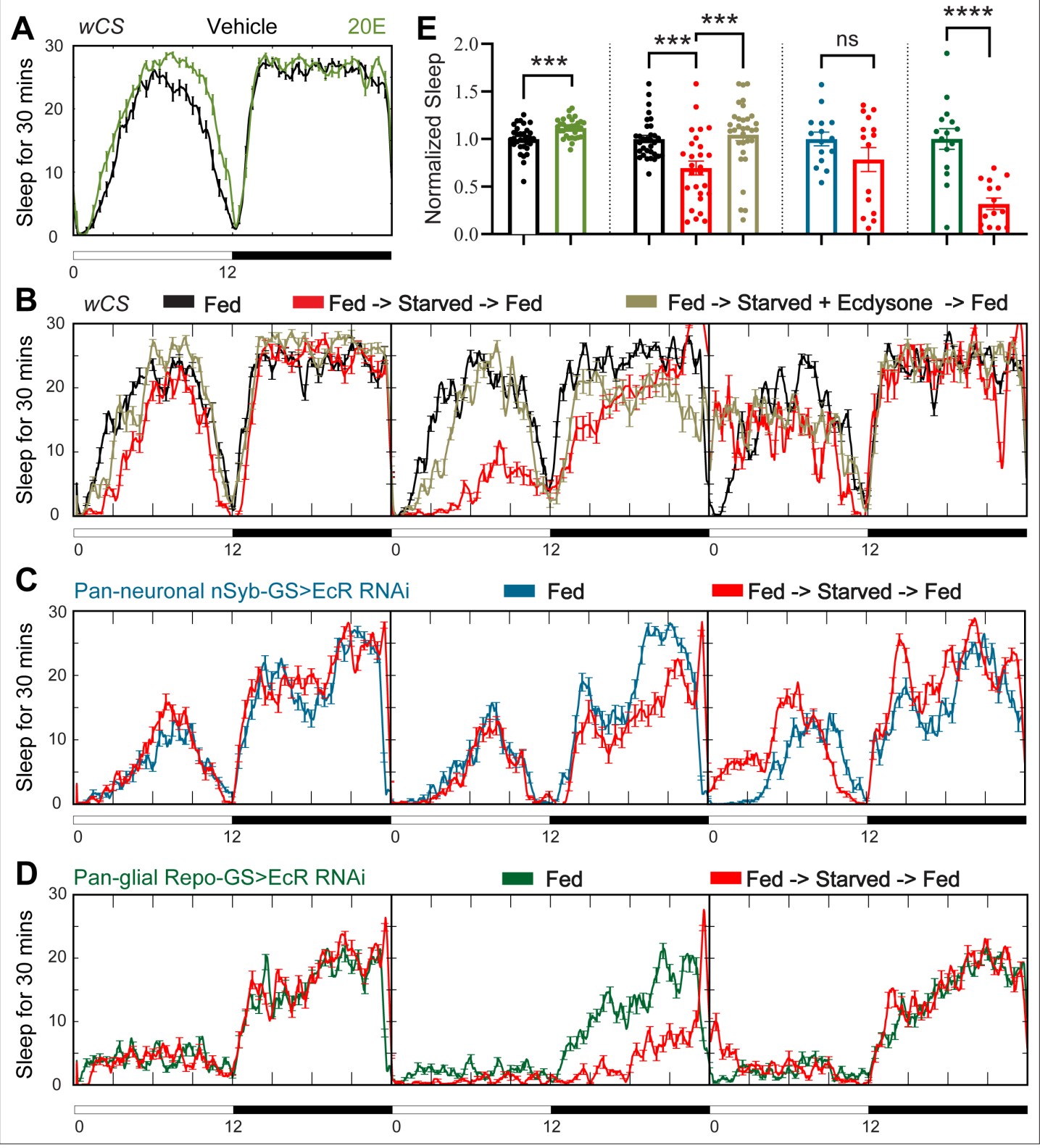

**Figure 3.** Ecdysone feeding prevents sleep loss in response to starvation, while ecdysone receptor (EcR) KD exacerbates starvation-induced sleep loss. Sleep was monitored for 3 days either under continuous feeding with and without 0.2 mM ecdysone (**A**) or under each of the following three conditions—continuous feeding for 3 days or with an intervention on the second day, which consisted of either starvation or starvation accompanied with the feeding of 0.2 mM ecdysone (**B**). The assay was conducted in (**A, B**) wild-type flies (*wCS*) and (**C, D**) nSyb-GS and Repo-GS EcR RNAi #1 flies. *N* = 15–32 for each genotype, and the experiment was repeated two times. Quantification of sleep nomalized to vehicle control group for the average

*Figure 3 continued on next page*

*Figure 3 continued*

daily sleep for (**A**) and ZT0–ZT23h interval of the second day for (**B–D**) is shown in (**E**). ZT0–ZT23 sleep data are shown as flies were flipped back to locomotor tubes with food during the last hour (ZT23–24). Bar graphs show mean ± standard error of the mean (SEM) and ns = not significant, p > 0.05, ***p < 0 .001, ****p < 0.0001. p values for comparisons between two groups were based on the Mann–Whitney test, and p values for comparison between three groups were calculated by one-way analysis of variance (ANOVA) with Tukey post hoc test. See also *Figure 3—source data 1*.

The online version of this article includes the following source data and figure supplement(s) for figure 3:

**Source data 1.** Sleep phenotypes of starvation on wild-types and ecdysone receptor (EcR) disrupted flies.

**Source data 2.** Sleep phenotypes of ecdysone treatment and heat shock when ecdysone receptor (EcR) was disrupted in neurons and glial cells.

**Source data 3.** Ecdysone levels in both fly brain and periphery, and the expression levels of ecdysone responsive genes.

**Figure supplement 1.** Ecdysone receptor (EcR) disruption in neurons or glia does not affect the response to heat shock.

**Figure supplement 2.** Ecdysone levels are higher in the fly periphery at ZT14 compared to ZT2.

signaling in the response to starvation, it could also be due to variability in the data. On the other hand, robust sleep loss was seen with starvation when EcR was knockdown in glia even though baseline sleep is already quite low in these flies (*Figure 3D, E*). Although the percentage of sleep loss was higher in Repo-GS>EcR RNAi flies than in controls (68% vs. 30%), because the baseline was lower, the absolute amount of sleep loss was about the same. These data indicate that starvation and ecdysone signaling in glia independently regulate sleep and a large part of sleep remaining in starved flies derives from ecdysone signaling in glia. We also examined the response to a stressor that increases sleep, heat shock, and found that sleep induced by heat shock was not affected by EcR knockdown in neurons or glia (*Figure 3—figure supplement 1B*).

## Ecdysone cycles in the periphery and has higher action on the brain at night

Given that ecdysone modulates rhythmic behavior, the question arises whether it is under circadian regulation. We used multiple methods, including a genetic reporter (hs-Gal4-EcR.LBD), mass spectrometry, and an ecdysone ELISA (enzyme-linked immunosorbent assay) kit, to determine if 20E cycles in the brain over a day–night cycle (*Ishimoto and Kitamoto, 2010*). However, ELISA analysis indicated that 20E level is significantly higher at ZT14 than ZT2 in the fly periphery but not in the fly brain (*Figure 3—figure supplement 2A*). In the fly brain, even genetic reporters and mass spectrometry approaches failed to detect any significant differences in 20E levels. Our inability to detect a cycle in the brain, despite a previously reported peak at ZT12 in the fly head (*Ishimoto and Kitamoto, 2010*), could reflect the rapid metabolism of 20E following its action on brain tissues. Alternatively, changes of 20E levels in the fly brain may be too small to be detected (*Vafopoulou and Steel, 2012*). In support of circadian regulation of ecdysone signaling, mRNA levels of ecdysone responsive E75 isoforms cycle with a peak at ZT12 in the fly head, suggesting that ecdysone acts rhythmically in the fly brain (*Figure 3—figure supplement 2B*), though this rhythmicity could also be contributed by direct clock regulation of E75 as suggested by *Abruzzi et al., 2011*. Given that E75 isoforms do not peak at ZT12 in the fly periphery, we suggest that elevated ecdysone at this time enters the brain instead of acting on peripheral tissues. To determine whether ecdysone enters the brain more efficiently at night, we examined whether E75 isoforms show differential responses to peripheral injection of 20E at ZT 6 versus ZT18. We normalized the change in E75 expression in the brain to the change in the body to account for inconsistency of injection and found that the mRNA levels of both E75A and E75B increased more at ZT18, compared to ZT6, after 20E injection, E75A increased 1.67-fold at ZT6, and 3.91 at ZT18, while E75B increased 17-fold at ZT6, and 51.92-fold at ZT18 (*Figure 3—figure supplement 2C*). This result indicates that peripheral ecdysone has higher action in the brain at night.

## EcR functions in cortex glia to affect sleep

Given the relevance of glia to EcR effects on sleep, we focused on uncovering its function in glial cells. We first assayed the expression of EcR in adult brains. Consistent with previous reports, antibody staining in the adult fly brain showed that EcR is broadly expressed, but largely in the cortex glia. To confirm the localization to cortex glia, we used two separate cortex glia drivers, *GMR77A03*-Gal4 and *Np2222*-Gal4, to drive the expression of mCD8-RFP. Co-staining these brains with an antibody against EcR confirmed that EcR is expressed primarily in the cortex glia (*Figure 4A*). When EcR was

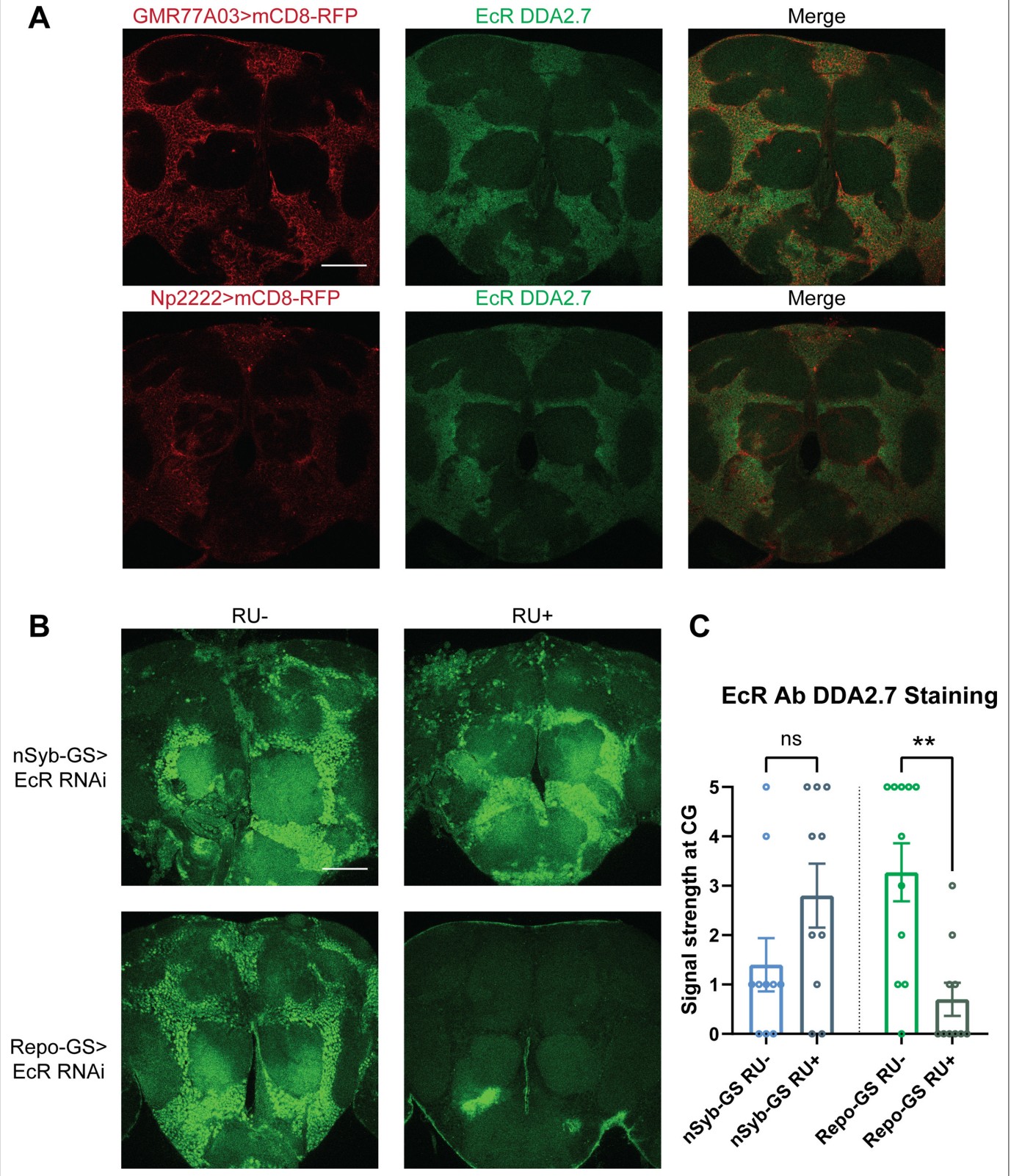

**Figure 4.** Ecdysone receptor (EcR) is expressed in cortex glia. (**A**) EcR antibody DDA2.7 staining overlaps with reporter expression driven by cortex glia drivers (GMR77A03>mCD8 RFP and Np2222>mCD8 RFP), indicating that EcR is expressed in the cortex glia. EcR antibody DDA2.7 staining also shows that EcR can be nuclear or cytoplasmic. (**B**) EcR antibody DDA2.7 staining is preserved and still observed in the nSyb-GS>EcR RNAi flies but is almost eliminated in the Repo-GS>EcR RNAi flies compared with the vehicle control flies. *N* = 10 per group. Scale bar: 100 μm. (**C**) Quantification of antibody

*Figure 4 continued on next page*

*Figure 4 continued*

staining in the fly brains' cortex glia layer, see also *Figure 4—source data 1*. Signal strength was scored manually based on the fluorescence intensity in the cortex glia region. Repo-GS>EcR RNAi #1 flies have significantly reduced staining based on an unpaired parametric Student's *t*-test. Bar graphs show mean ± standard error of the mean (SEM) and ns = not significant, p > 0.05, **p < 0.01.

The online version of this article includes the following source data for figure 4:

**Source data 1.** Quantification of Anti-EcR staining in the cortex glia layer.

knocked down in glia with Repo-GS, EcR antibody staining was greatly diminished. However, signals were preserved when EcR was knocked down in neurons, leading us to infer that glia, in particular cortex glia, are the major site of expression (*Figure 4B*). Although not evident in these images, EcR expression was also detected in surface glia of the blood–brain barrier (BBB) (data not shown).

To determine whether sleep phenotypes correlate with the expression pattern of EcR, we then knocked down EcR constitutively in different glial subpopulations by crossing subglial Gal4 lines with the EcR RNAi #1 line that targets all EcR isoforms. However, EcR knockdown with most of these Gal4 lines caused failure of eclosion, indicating the importance of EcR in glial cells during development. Three Gal4 lines produced viable adult flies: one cortex glia-specific driver GMR77A03, one astrocyte-like glia-specific driver Eaat-1, and one ensheathing glia-specific driver GMR56F03. Of these three Gal4 lines, sleep was only reduced when EcR was knocked down using the cortex glia-specific driver (*Figure 5A, C*).

To avoid issues of lethality and to distinguish between developmental and adult-specific roles, we used the TARGET system to repress Gal4 transcription in the developmental stage, after which the flies were transferred to a (restrictive) elevated temperature to degrade temperature-sensitive GAL80$^{ts}$ and allow expression of Gal4 (*Figure 5D*, *McGuire et al., 2004*). At the (permissive) low temperature, when TUB-GAL80ts blocks Gal4 expression, flies expressing EcR RNAi flies with the cortex glia driver NP2222 trended toward reduced sleep but sleep decreased significantly at the restrictive temperature when RNAi expression was activated to knock down EcR in cortex glia (*Figure 5E, G*). Knockdown of EcR with the 9-137 Gal4 driver, which targets all glia of the BBB, did not affect sleep (*Figure 5F, H*), and nor did knockdown of EcR in ensheathing or astrocyte glia (*Figure 5—figure supplement 1*), suggesting that effects of EcR on sleep are restricted to specific glia, with cortex glia being the primary physiological site of action.

We also overexpressed EcR using the same set of subglia drivers and found that only flies expressing EcR with the 9-137 Gal4 driver had elevated sleep (*Figure 5B*). These flies were also resistant to sleep deprivation, indicating enhanced sleep need. Restricting overexpression to the adult stage with TubGal80 did not affect sleep, suggesting that the sleep-promoting effect observed in 9-137 Gal4 flies is developmental (*Figure 5—figure supplement 1D*).

## Ecdysone modulates sleep by mobilizing lipid droplets in glial cells

The cortex glia, especially the superficial cortex glial cells, are enriched in lipid droplets (LDs) in the third instar larval stage, and ecdysone signaling can mobilize these lipids when needed (*Kis et al., 2015*). To determine whether ecdysone in adult flies has a similar effect on lipids, we assayed LDs in glial cells of flies fed ecdysone. We measured LDs in the fly brain by BODIPY staining and found that LDs, which accumulate mostly in the cortex and ensheathing glia, were significantly smaller in flies fed ecdysone, relative to control groups, at Zeitgeber Time (ZT)12. At the same time, the total count was not affected (*Figure 6A–D*). Given that ecdysone feeding mobilizes LDs, we speculated that glial EcR knockdown might cause flies to accumulate more lipids. We performed lipid staining, comparing vehicle controls and RU486-induced groups of Repo-GS>EcR RNAi flies (*Figure 6—figure supplement 1A–D*), and found that EcR knockdown in glial cells resulted in more LDs, RU treatment of UAS control flies had no effect, together suggesting that ecdysone signaling bidirectionally affects LDs in glial cells. Increased sleep with ecdysone feeding reduces LDs while reduced sleep with loss of ecdysone signaling increases LDs.

We next asked if this effect on lipids was relevant for sleep induction by ecdysone by measuring the ecdysone response of a lipid storage mutant (*Kamoshida et al., 2012*). The lipid storage droplet 2 (LSD2) protein modulates lipid accumulation and response to starvation, so *lsd2* mutants are lean and sensitive to starvation (*Thimgan et al., 2010*). We found that ecdysone does not promote sleep in *lsd2* mutants (*Figure 6E*). However, feeding *lsd-2* mutant flies gaboxadol, a GABA agonist known

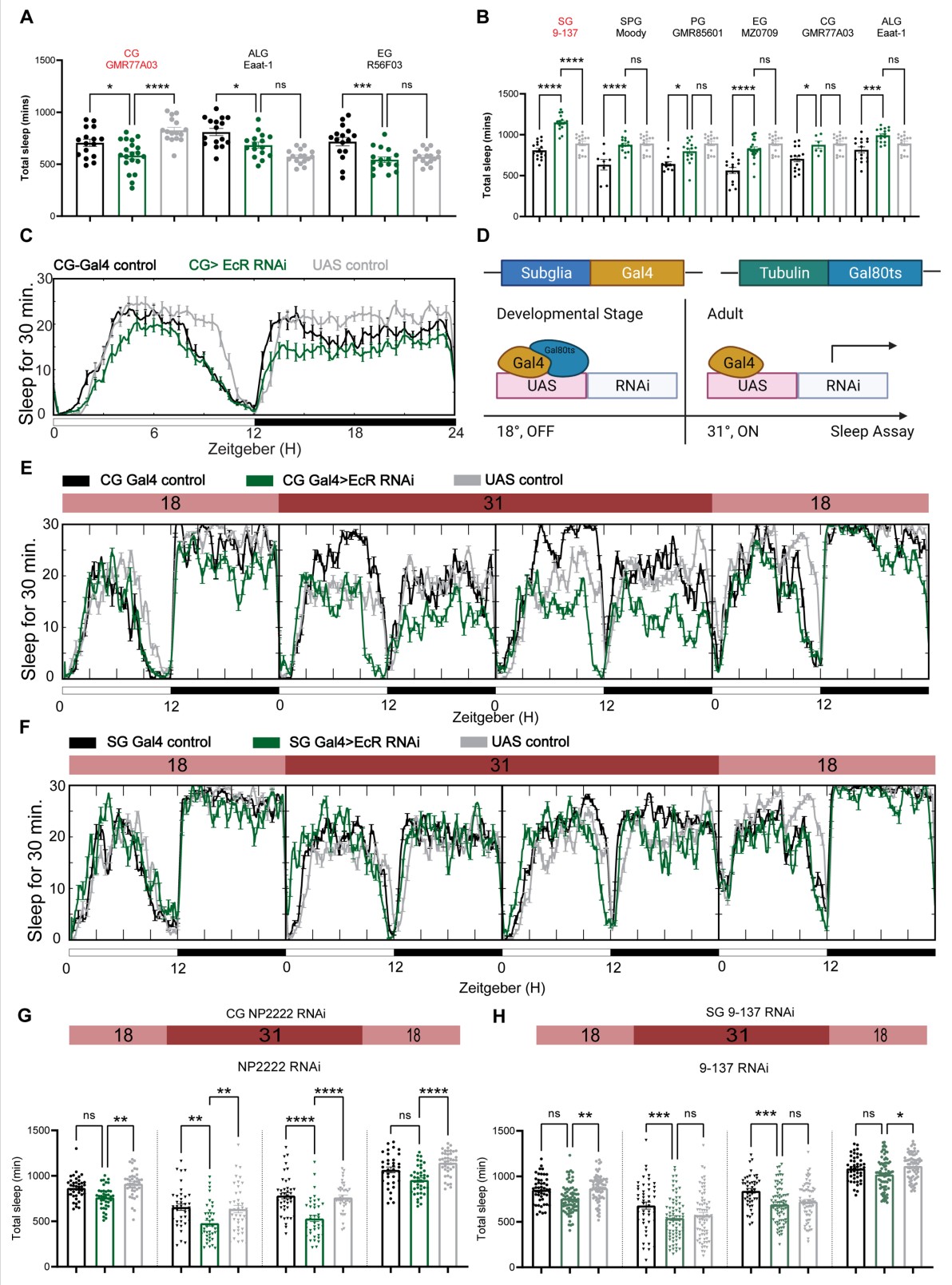

**Figure 5.** Ecdysone receptor (EcR) functions in cortex glia to affect sleep. (**A**) Multiple constitutive Gal4 drivers labeling different subglial populations were crossed to EcR RNAi flies. Only GMR77A03 for cortex glia (CG), Eaat-1 for astrocyte-like glia (ALG), and GMR56F03 for ensheathing glia (EG) drivers produced viable adult progeny flies, and only GMR77A03>EcR RNAi #1 flies showed reduced total sleep compared with control flies. Green chart columns are experimental groups, and neighboring black and gray columns are Gal4 and UAS flies controls, respectively. *N* = 16–20 per genotype.

*Figure 5 continued on next page*

Figure 5 continued

(**B**) Overexpression of EcR by subglial Gal4 drivers—9-137 Gal4 to drive expression in the surface glia (SG), Moody-Gal4 for subperineurial glia (SPG), GMR85G01 for perineurial glia (PG), MZ008-Gal4 for ensheathing glia (EG), GMR77A03 for cortex glia (CG), and Eaat-1 for astrocyte-like glia (ALG). Only overexpression of EcR in the surface glia promotes sleep. *N* = 16–24 per genotype. (**C**) Representative sleep traces of the cortex glia GMR77A03>EcR RNAi #1 flies. (**D**) Gal4/Tubulin-gal80ts was used to achieve adult-specific knockdown of EcR in different subglial populations. Under permissive temperature, Gal80ts inhibits Gal4 activation of UAS, but under restrictive temperature, Gal80ts is inactivated, and genes under the regulation of UAS are expressed. (**E, F**) Sleep traces resulting from EcR knockdown in the cortex glia using NP2222-Gal4/tubulinGal80ts and surface glia using 9-137 Gal4/tubulinGal80ts. F1 progeny flies were kept at 18 degrees for 1 day, and then the temperature was switched to 31 degrees to inactivate the Gal80ts and thus achieve knockdown of EcR over the following 2 days. Subsequently, temperatures were decreased back to 18 degrees. (**G, H**) show quantification of total sleep of all EcR knockdown flies in the cortex glia using NP2222-Gal4/tubulinGal80ts and surface glia using 9-137 Gal4/tubulinGal80ts, *N* = 34–77 per genotype. Total sleep of each genotype was calculated and compared to controls for the above 4 days. Bar graphs show mean ± standard error of the mean (SEM), ns = not significant, $p > 0.05$, *$p < 0.05$, **$p < 0.01$, ***$p < 0.001$, ****$p < 0.0001$. p values for each comparison were calculated by one-way analysis of variance (ANOVA) with Tukey post hoc test. See also *Figure 5—source data 1*.

The online version of this article includes the following source data and figure supplement(s) for figure 5:

**Source data 1.** Sleep phenotypes resulting from subglial knockdown of ecdysone receptor (EcR).

**Source data 2.** Sleep phenotypes of adult-specific ecdysone receptor (EcR) disruption in different subglial populations.

**Figure supplement 1.** Adult-specific disruption of ecdysone receptor (EcR) in most subglial populations does not affect sleep.

to induce deep sleep, significantly increased sleep (*Figure 6—figure supplement 2*), indicating that non-responsiveness to ecdysone was not due to an inability of *lsd-2* flies to respond to sleep-promoting interventions. Lack of a sleep-promoting effect of ecdysone on *lsd-2* mutants, together with bidirectional effects of ecdysone on LDs, indicate strongly that ecdysone affects sleep in large part by modulating LDs in glial cells, especially cortex glia (*Figure 6—figure supplement 3*).

## Discussion

While sleep traditionally has been regarded as a neuronally driven behavior, glial cells can regulate sleep by affecting neuronal activity and possibly even mediate functions attributed to sleep, such as waste clearance, nutrient transfer, and repair (*Cai et al., 2021*; *Donlea et al., 2014*; *Keene et al., 2010*; *Chi-Castañeda and Ortega, 2016*; *Seidner et al., 2015*; *Singh and Donlea, 2020*; *Stanhope et al., 2020*; *Yildirim et al., 2019*). We show here that glia are also a major target of steroid hormone signaling to regulate sleep. In addition, our data support an important role for lipid metabolism in controlling sleep.

In *Drosophila*, ecdysone is the major steroid hormone, and it plays an essential role in regulating metamorphosis and molting (*Yamanaka et al., 2013*). Ecdysone also affects the development of the nervous system and neuronal remodeling, processes that may involve glial function (*Yu and Schuldiner, 2014*). During development, ecdysone is synthesized from dietary cholesterol in the prothoracic gland and in other peripheral tissues that control molting and eclosion in larvae and pupae, but it declines during pupa-adult transitions and remains low in adults (*Yamanaka et al., 2013*; *Kannangara et al., 2021*). Nevertheless, it is implicated in adult functions such as memory formation and stress resistance (*Ishimoto and Kitamoto, 2011*). Ecdysone was also shown to affect sleep (*Ishimoto and Kitamoto, 2011*), raising questions of the mechanism by which it does so and the cells on which it acts in adults. Our finding that ecdysone regulates sleep through metabolic mechanisms in glia is consistent with other metabolic functions attributed to it; for instance, it controls developmental transitions in response to nutrient signals (*Christensen et al., 2020*; *Lee et al., 2000*; *Uyehara and McKay, 2019*; *Xu et al., 2020*; *Yamanaka et al., 2013*). Indeed, ecdysone may affect the whole-body lipid profile to modulate physiology and behavior (*Sieber and Spradling, 2015*).

The gonads, in particular the ovaries in females, are thought to be the major source of the ecdysone in adults (*Ahmed et al., 2020*). However, several other tissues also express ecdysone biosynthesis genes (*Li et al., 2022*), so multiple peripheral tissues could contribute to circulating ecdysone levels. Consistent with this notion, our efforts to observe sleep changes by disrupting the synthesis of ecdysone in any one of the following peripheral tissues—gut, fat body, and ovary—failed (data not shown). We speculate that knockdown of biosynthetic enzymes in any one peripheral tissue does not have a significant effect because ecdysone can still be derived from other tissues. Notably, knockdown in the brain did not have any effect either. Also, Halloween genes have very little to no expression in

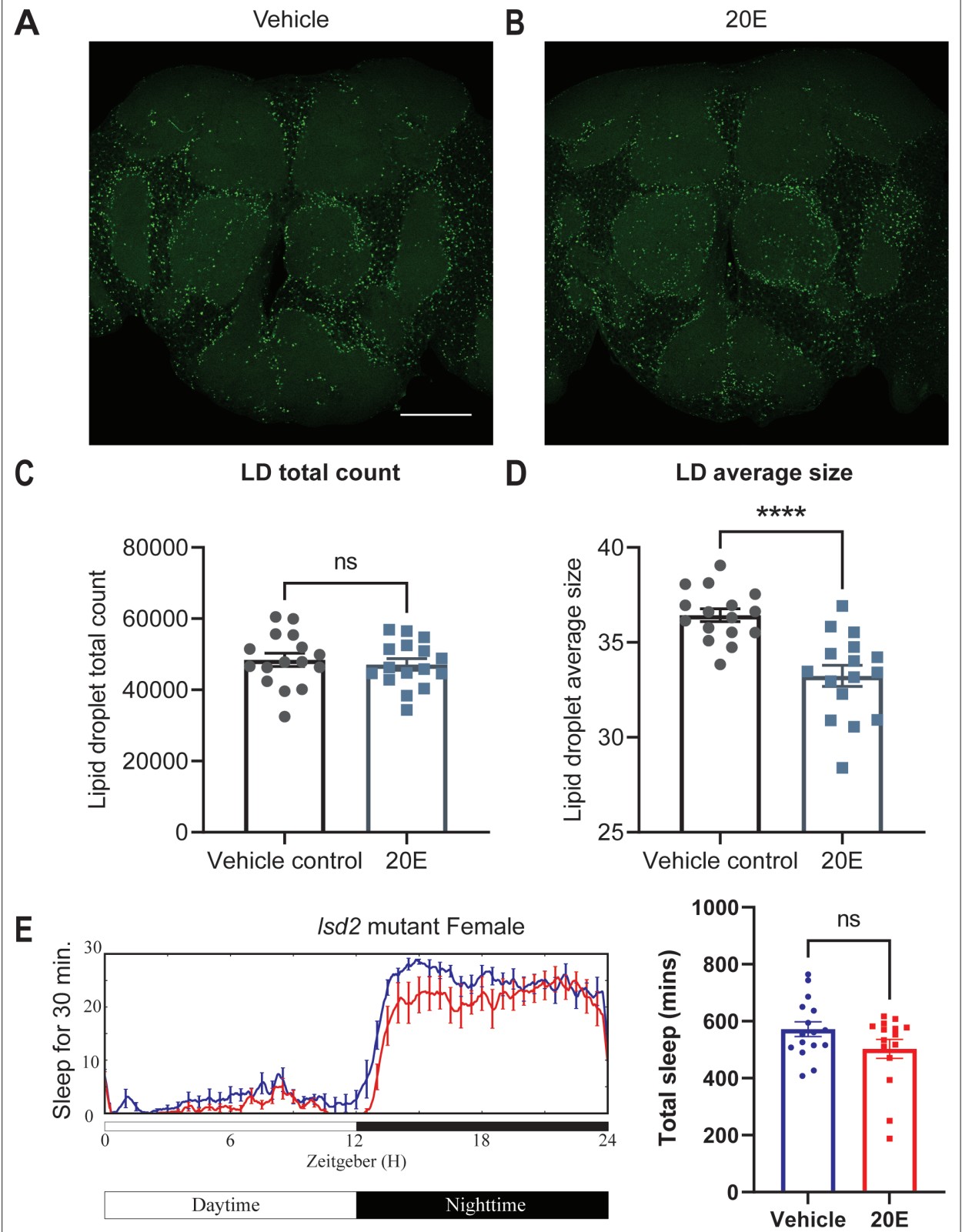

**Figure 6.** Lipid metabolism mediates the effects of 20E on sleep. (**A, B**) Lipid droplet (LD) staining of representative brains from flies treated with vehicle or 0.5 mM 20E. A z-stack slice that shows the maximal structure of the cortex glia was selected, and brightness and contrast were auto-adjusted by ImageJ for better visualization. LDs are stained by the lipophilic dye BODIPY 493. (**C, D**) 0.5 mM 20E treatment does not significantly affect the total count of LDs but likely mobilizes lipids to lead to smaller LDs. LD count and size were analyzed and calculated using ImageJ as detailed in the Materials

*Figure 6 continued on next page*

*Figure 6 continued*

and methods. *N* = 16 per group from two independent repeats. Statistical comparisons used unpaired parametric Student's *t*-test. Bar graphs show mean ± standard error of the mean (SEM) and ns = not significant, p > 0.05, ****p < 0.0001. (**E**) A representative sleep trace of *lsd2* mutant flies with vehicle control or 0.2 mM ecdysone. As previously reported, total sleep increases with 20E treatment in *w*CS,but the sleep-promoting effect of the 20E is lost in the *lsd2* mutant flies, *N* = 16 per group from two separate experiments. Statistical comparisons used unpaired parametric Student's *t*-test. See also *Figure 6—source data 1*.

The online version of this article includes the following source data and figure supplement(s) for figure 6:

**Source data 1.** The effects of ecdysone on lipid droplets and sleep.

**Source data 2.** Lipid droplets changes resulting from glial ecdysone receptor (EcR) knockdown.

**Source data 3.** Sleep phenotypes of gaboxadol treatment in *lsd-2* mutant flies.

**Figure supplement 1.** EcR knockdown in glial cells resulted in more LDs.

**Figure supplement 2.** Gaboxadol promotes sleep in *lsd-2* mutant flies.

**Figure supplement 3.** Model for sleep function of ecdysone in cortex glia.

the fly brain based on fly brain single-cell sequencing (*Davie et al., 2018*), suggesting that ecdysone or 20E derives from the periphery to modulate sleep via glia.

In *Drosophila*, around 10–15% of cells in the brain are glial cells belonging to one of five distinct groups: cortex glia, astrocyte-like glia, ensheathing glia, perineurial, and subperineural glia (*Edwards and Meinertzhagen, 2010*; *Yildirim et al., 2019*). Perineurial and subperineural glia together serve as the BBB regulating permeability, controlled by the circadian system (*Zhang et al., 2018*). BBB glia also have a role in sleep, such that endocytosis in these cells increases during sleep and depends upon the prior duration of wakefulness (*Artiushin et al., 2018*). Astrocyte-like glia have a distinctive shape that allows remarkably close physical contact with synapses and is thought to be important for the clearance of neurotransmitters in the synaptic space (*Freeman, 2015*). In support of this, arylalkyl-amine *N*-acetyltransferase 1 (AANAT1), which acetylates and inactivates monoamines, acts in astrocytes to affect sleep (*Davla et al., 2020*). Also, calcium signaling in astrocytes appears to contribute to sleep need (*Blum et al., 2021*), and astrocytic GABA transporter maintains proper GABA tone in specific wake-promoting circadian neurons (*Chaturvedi et al., 2022*). Cortex glia cells encapsulate neuronal cell bodies and provide nutrients to neurons (*Doherty et al., 2009*). Cortex and ensheathing glia accumulate most LDs in the brain, and fatty acid-binding protein (Fabp), which promotes sleep, is expressed in both glial populations (*Gerstner et al., 2011*). In addition, ensheathing glia regulate both sleep and metabolic rate via the taurine transporter (*Stahl et al., 2018*). Thus, roles of cortex and ensheathing glia in sleep are likely linked to their function in metabolic, which includes nutrient transfer and balance between neurons and glia (*Bittern et al., 2021*). We note though that the effects of Amyloid precursor protein (App), which regulates the production and deposition of toxic amyloid peptides, on sleep are mediated by cortex glia (*Farca Luna et al., 2017*).

We show here that the ecdysone signaling pathway functions in adult cortex glia and neurons to affect circadian locomotor rhythms and sleep. The specific cellular targets through which ecdysone regulates sleep were previously not known. Additionally, while ecdysone can regulate circadian rhythms through neuronally expressed receptors (*Kumar et al., 2014*), we find that knockdown of glial EcRs results in more severe behavioral arrhythmicity than neuronal knockdown (*Figure 2I–K*). As *Drosophila* astrocytes regulate circadian locomotor rhythms (*Ng et al., 2011*), they are attractive candidates for mediating the effects of ecdysone on rhythms. Our finding that EcR in specific subglial cells is also vital to the development of flies is surprising even though NHRs have well-documented roles in development. Ecdysone treatment upregulates the gene *glial cell missing (gcm)*, while knockdown of EcR reduces the expression of *gcm* in vitro, suggesting that ecdysone influences the development and morphology of glial cells by regulating *gcm*, which determines the fate of the lateral glial cells (*Wang et al., 2014*). Thus, glia are an important target for biological actions of ecdysone on development and adult behavior.

We show that exogenous ecdysone mobilizes lipids accumulated in cortex glia to promote sleep, and glial EcR knockdown results in the accumulation of LDs. Since cortex glia cells surround and compartmentalize neuronal cell bodies, they may modulate neuronal activity by facilitating metabolite transfer to neurons. Aside from cortex glia, the only glial type to yield a phenotype with manipulations of ecdysone signaling is the surface glia. The sleep phenotype by EcR overexpression in surface

glia may be non-physiological as a strong Gal4 line produces it, but it nevertheless suggests that the surface glia can support ecdysone signaling.

Our finding that lipid metabolism is important for ecdysone-induced sleep fits with increasing evidence of interactions between sleep and lipids. Loss of sleep alters the lipid profile across species, including in human peripheral blood (*Davies et al., 2014*; *Hinard et al., 2012*; *Weljie et al., 2015*). Conversely, lipids have been shown to regulate sleep; for instance, the *lsd2* mutant we used here, as well as a mutant lacking a lipase, affect rebound after sleep deprivation in *Drosophila* (*Thimgan et al., 2010*). Interestingly, specific lipids are also implicated as secreted sleep inducers (somnogens) that promote sleep following deprivation in mammals (*Cravatt et al., 1995*). Studies in worm also showed that sleep is associated with fat mobilization, and deficits in energy mobilization in sensory neuroendocrine cells cause sleep defects (*Grubbs et al., 2020*).

How ecdysone synthesis is regulated and how its function is integrated with innate and environmental changes needs further study. Juvenile hormone (JH), which works together with ecdysone during developmental stages, differentially affects sleep in male and female flies (*Leinwand and Scott, 2021*; *Schwedes and Carney, 2012*; *Wu et al., 2018*); it is reasonable to hypothesize that JH interacts with ecdysone in the context of sleep. The human ortholog of E75 is Reverb, which has broad effects on metabolism in peripheral tissues based on work in mice and humans, so the functions of EcR/E75 in peripheral tissues may also be linked to metabolism (*Ding et al., 2021*; *Zhang et al., 2017*). In addition, Reverb has high expression in mouse brain glial cells, where it could function to affect mouse sleep (*Chi-Castañeda and Ortega, 2017*).

In summary, the endocrine system and the circuitry underlying circadian rhythms and sleep are intertwined during developmental stages (*Aldabal and Bahammam, 2011*; *Gamble et al., 2014*; *Morgan and Tsai, 2015*), and we now find that ecdysone acts through specific glial cells to affect circadian rhythms and sleep in adults. Similar effects of the ecdysone downstream target E75 and the ecdysone importer implicate canonical nuclear hormone signaling in sleep regulation. The mechanism involves LD mobilization, emphasizing the importance of glia and lipid metabolism in sleep regulation. Our findings are likely just the tip of the iceberg concerning endocrine regulation of sleep. We expect this to be a rich area of investigation in the future, as peripheral effects on brain function are increasingly recognized.

## Materials and methods
See Appendix 1.

## Contact for reagent and resource sharing
Further information and requests for resources and reagents should be directed to lead contact Amita Sehgal (amita@pennmedicine.upenn.edu).

## Experimental model and subject details
### Fly stock and maintenance
*Drosophila* stocks were obtained from our lab stock, Bloomington *Drosophila* stock center (BDSC), or the Vienna *Drosophila* Resource Center (VDRC). The white-CantonS (*w*CS) strain was used as wild-type unless specified. The genotype information of the flies used in each experiment is listed in the Key resource table. Strains used from the lab stocks include nSyb-GeneSwitch, Repo-GeneSwitch, all the subglial lines, and four subglial tubGal80ts lines. nSyb-GeneSwitch, Repo-GeneSwitch experiments were conducted with UAS-Dcr2 to promote RNAi efficiency. EcR RNAi lines are from the lab stock originally purchased from VDRC. The *lsd-2* flies were gifts from Dr. Michael Welts lab, and EcI lines were gifts from Dr. Naoki Yamanaka lab.

## Behavior measurement in *Drosophila*
Flies were raised on cornmeal-molasses medium under 12:12 hr light:dark cycle at 25°C unless specified otherwise. Twelve female virgins and four male flies were usually crossed together to generate different flies of genotypes. Parental flies were cleared after seven days, and F1 progeny collected after 12 days. On day17, 5- to 7-day-old female flies were loaded into locomotor tubes for behavior tests as previously described (*Davla et al., 2020*). Locomotor tubes were 60-mm glass tubes, waxed

and loaded with 2% agar containing 5% sucrose as fly food on one side, and yarn on the other side to restrain the behavior of flies inside the glass tubes. For GeneSwitch experiments, 0.5 mM RU-486 (mifepristone) was added to the fly food to activate the GeneSwitch. Three constitutive days' data were used for sleep analysis by Pysolo (https://www.pysolo.net/) (Gilestro and Cirelli, 2009), and seven constitutive days' data under constant dark were used for circadian rhythm analysis by ClockLab (https://actimetrics.com/products/clocklab/).

For starvation assay, starvation tubes were made with only 2% agarose as fly food. Five- to seven-day-old flies were first loaded into normal locomotor tubes for days 0 and 1, then they were transferred to starvation tubes between ZT23 and ZT0 on day 2. Flies were monitored for sleep in starvation tubes for 1 or 2 days and then transferred back to normal tubes at ZT23–ZT0. Post-starvation sleep was measured for 1 day.

## Ecdysone treatment, immunohistochemistry, and imaging

Ecdysone was purchased in powder form and was dissolved in ethanol. Then it was mixed with 2% agarose containing 5% sucrose to make the different doses of ecdysone tubes, and the same amount of ethanol as in the ecdysone solution was used to make the vehicle control tubes. Only 2% agarose and ecdysone were used to make ecdysone starvation tubes. Five- to seven-day-old flies were loaded into locomotor tubes for ecdysone treatment and behavior recording to verify that ecdysone promotes sleep in ecdysone-treated flies. For ecdysone feeding/starvation assay, all flies were kept in normal tube on days 0, 1, and 2. Flies were then transferred to either starvation tubes, or ecdysone starvation tubes between ZT23 and ZT0 on day 2. Flies were kept in these ecdysone/starvation tubes for 1 day and transferred back to normal tubes again between ZT23 and ZT0 on day 3 and recorded for another day. Data from days 2 to 4 were used for sleep analysis. For gaboxadol hydrochloride treatment, it was dissolved in water and diluted to 0.1 mg/ml as final concentration for locomotor tube experiments. Sleep was measurement from days 2 to 4, after flies were loaded into gaboxadol tubes at day 0. For TARGET system experiments, flies were raised at permissive temperature 18°C, and 5- to 7-day-old adult flies were used to loaded and were kept at 18°C from days 0 to 2, 31°C from days 3 to 4, and 18°C for day 5. Data from days 2 to 5 were collected for sleep analysis.

BODIPY493 was used for brain LD staining. Flies were loaded into locomotor tubes and subjected to 0.5 mM ecdysone treatment for 12 or 36 hr. Both control and ecdysone-treated flies were then dissected in the phosphate-buffered saline (PBS), fixed in 4% Paraformaldehyde (PFA) solution for 20 min, and washed three times with PBS + 0.3% Triton (PBT). Then brains were left in PBT at 4 degrees overnight and transferred to 1 µg/ml BODIPY493 in PBT for 20 min. Later they were mounted for imaging.

Brains were imaged with the oil-immersion ×40 lens of a confocal microscope at a resolution of 1024 × 1024. Raw images were processed with FIJI ImageJ. The first step was to remove non-lipid trash and exclude artificial signals or signals outside the brain area. The second step was to quantify the LD count, area, and total brain tissue area. Later, the LDs count will be normalized by the brain tissue area, and LD size calculated by LD area divided by total count. Quantifications were conducted by using ImageJ Macro.

## Ecdysone assay and real-time PCR

20E measurement was performed with the 20-hydroxyecdysone EIA kit (Cayman, item No. 501390). 5–10 fly bodies or 10–15 brains were dissected and homogenized in 70% methanol. The homogenates were then dried by the evaporator, redissolved in the kit's buffer, and measured based on the manufacturer's protocol. Adult fly brains (usually 10) or fly bodies (usually 5) were subjected to RNeasy Plus Mini Kit for RNA extraction, followed by cDNA reverse transcription using random hexamers and Superscript II (Invitrogen) to generate cDNA. The cDNA was then amplified using SYBR Green PCR mix (Cat# 4364346) and oligonucleotides listed in the Key resource table with the Applied Biosystems ViiA7 qPCR machine. Relative transcript levels were calculated by ddCT.

## Statistical analysis

GraphPad Prism was used for all statistical tests. Data were tested for normality using the D'Agostino–Pearson test and Shapiro–Wilk test. Normally distributed data were then tested with an unpaired parametric Student's t-test for two independent groups and a one-way analysis of variance with Tukey

post hoc test for multiple independent groups. Non-normally distributed data, such as sleep bout numbers, which are usually non-normally distributed, were analyzed with a nonparametric test like the Mann–Whitney test for two samples and the Kruskal–Wallis test with Dunn's multiple comparisons test for three samples or above. Statistic tests used for each experiment are indicated in the figure legends, and data are presented as means and standard error of the mean.

## Acknowledgements

This work was supported by Howard Hughes Medical Institute (HHMI). We thank Anna Kolesnik, Joy Shon, Xanthe Heifetz Ament, Samantha Killiany, and Kiet Luu for continuing technical assistance and all other members of the Sehgal lab for feedback, comments, and reagents. The schematic was created with https://biorender.com/. We thank Dr. Michael Welts and Dr. Naoki Yamanaka for generously sharing fly lines with us.

## Additional information

### Funding

| Funder | Grant reference number | Author |
|---|---|---|
| Howard Hughes Medical Institute | | Amita Sehgal |
| National Institute of Neurological Disorders and Stroke | R01NS048471 | Amita Sehgal |
| National Institutes of Health | R01DK120757 | Amita Sehgal |

The funders had no role in study design, data collection, and interpretation, or the decision to submit the work for publication.

### Author contributions

Yongjun Li, Conceptualization, Resources, Data curation, Software, Formal analysis, Supervision, Validation, Investigation, Visualization, Methodology, Writing – original draft, Project administration, Writing – review and editing; Paula Haynes, Resources, Data curation, Software, Formal analysis, Methodology, Writing – review and editing; Shirley L Zhang, Resources, Software, Formal analysis, Supervision, Investigation, Methodology; Zhifeng Yue, Resources, Data curation, Methodology; Amita Sehgal, Conceptualization, Formal analysis, Supervision, Funding acquisition, Validation, Investigation, Project administration, Writing – review and editing

### Author ORCIDs

Yongjun Li ORCID http://orcid.org/0000-0002-3618-6041
Shirley L Zhang ORCID http://orcid.org/0000-0002-6672-2044
Amita Sehgal ORCID http://orcid.org/0000-0001-7354-9641

### Decision letter and Author response

Decision letter https://doi.org/10.7554/eLife.81723.sa1
Author response https://doi.org/10.7554/eLife.81723.sa2

## Additional files

### Supplementary files

• MDAR checklist

### Data availability

All data analyzed and reported in this study are included in the manuscript, supplementary tables, and source data linked to figures.

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

# Appendix 1

**Appendix 1—key resources table**

| Reagent type (species) or resource | Designation | Source or reference | Identifiers | Additional information |
|---|---|---|---|---|
| Genetic reagent (*D. melanogaster*) | w[*]; P{w[+mW.hs]=Switch2} GSG2326 | Bloomington Stock Center | BDSC:40990 | |
| Genetic reagent (*D. melanogaster*) | w[*]; P{y[+t7.7] w[+mC]=UAS-Eip78C.miRNA}attP16/CyO | Bloomington Stock Center | BDSC:44390 | |
| Genetic reagent (*D. melanogaster*) | w[*]; P{y[+t7.7] w[+mC]=UAS Hr3.miRNA} attP16 | Bloomington Stock Center | BDSC:44399 | |
| Genetic reagent (*D. melanogaster*) | w[*]; P{y[+t7.7] w[+mC]=UAS Hr96.miRNA} attP16 | Bloomington Stock Center | BDSC:44395 | |
| Genetic reagent (*D. melanogaster*) | w[*]; P{y[+t7.7] w[+mC]=UAS-Hnf4.miRNA}attP16/CyO | Bloomington Stock Center | BDSC:44398 | |
| Genetic reagent (*D. melanogaster*) | y[1] v[1]; P{y[+t7.7] v[+t1.8]=TRiP.JF02546}attP2 | Bloomington Stock Center | BDSC:27258 | |
| Genetic reagent (*D. melanogaster*) | y[1] sc[*] v[1]; P{y[+t7.7] v[+t1.8]=TRiP.HMS01620} attP2 | Bloomington Stock Center | BDSC:36729 | |
| Genetic reagent (*D. melanogaster*) | w[*]; P{y[+t7.7] w[+mC]=UAS Hr78.miRNA} attP16 | Bloomington Stock Center | BDSC:44393 | |
| Genetic reagent (*D. melanogaster*) | y[1] v[1]; P{y[+t7.7] v[+t1.8]=TRiP.JF02545}attP2 | Bloomington Stock Center | BDSC:27242 | |
| Genetic reagent (*D. melanogaster*) | y[1] sc[*] v[1]; P{y[+t7.7] v[+t1.8]=TRiP.HMS01316} attP2 | Bloomington Stock Center | BDSC:34329 | |
| Genetic reagent (*D. melanogaster*) | y[1] sc[*] v[1]; P{y[+t7.7] v[+t1.8]=TRiP.HMS01951} attP2 | Bloomington Stock Center | BDSC:39032 | |
| Genetic reagent (*D. melanogaster*) | y[1] v[1]; P{y[+t7.7] v[+t1.8]=TRiP.JF02537}attP2 | Bloomington Stock Center | BDSC:29373 | |
| Genetic reagent (*D. melanogaster*) | y[1] v[1]; P{y[+t7.7] v[+t1.8]=TRiP.HMS02272} attP40 | Bloomington Stock Center | BDSC:41707 | |
| Genetic reagent (*D. melanogaster*) | w[*]; P{y[+t7.7] w[+mC]=UAS Hr83.miRNA} attP16 | Bloomington Stock Center | BDSC:44397 | |
| Genetic reagent (*D. melanogaster*) | w[*]; P{y[+t7.7] w[+mC]=UAS svp.miRNA} attP16 | Bloomington Stock Center | BDSC:44394 | |
| Genetic reagent (*D. melanogaster*) | w[*]; P{y[+t7.7] w[+mC]=UAS ERR.miRNA} attP16/CyO | Bloomington Stock Center | BDSC:44391 | |
| Genetic reagent (*D. melanogaster*) | w[*]; P{y[+t7.7] w[+mC]=UAS Hr38.miRNA} attP16/CyO | Bloomington Stock Center | BDSC:44396 | |
| Genetic reagent (*D. melanogaster*) | y[1] v[1]; P{y[+t7.7] v[+t1.8]=TRiP.JF02738}attP2 | Bloomington Stock Center | BDSC:27659 | |
| Genetic reagent (*D. melanogaster*) | y[1] v[1]; P{y[+t7.7] v[+t1.8]=TRiP.HMS00019} attP2/TM3, Sb[1] | Bloomington Stock Center | BDSC:33625 | |

*Appendix 1 Continued on next page*

*Appendix 1 Continued*

| Reagent type (species) or resource | Designation | Source or reference | Identifiers | Additional information |
|---|---|---|---|---|
| Genetic reagent (*D. melanogaster*) | y[1] v[1]; P{y[+t7.7] v[+t1.8]=TRiP.HMS00018} attP2/TM3, Sb[1] | Bloomington Stock Center | BDSC:33624 | |
| Genetic reagent (*D. melanogaster*) | y[1] v[1]; P{y[+t7.7] v[+t1.8]=TRiP.JF02432}attP2 | Bloomington Stock Center | BDSC:27086 | |
| Genetic reagent (*D. melanogaster*) | w[*]; P{y[+t7.7] w[+mC]=UAS Hr4.miRNA} attP16 | Bloomington Stock Center | BDSC:44392 | |
| Genetic reagent (*D. melanogaster*) | y[1] w[*]; PBac{y[+mDint2] w[+mC]=I-SceI(FRT.Rab9-GAL4.ATG(loxP.3xP3-RFP))} VK00033/TM3, Sb[1] | Bloomington Stock Center | BDSC:51587 | |
| Genetic reagent (*D. melanogaster*) | w;; Repo-Gal4, 6 x crossed to Iso | Bloomington Stock Center | BDSC:7415 | |
| Genetic reagent (*D. melanogaster*) | w[1118]; UAS-EcR.A.dsRNA/ TM3 | Bloomington Stock Center | BDSC:9328 | |
| Genetic reagent (*D. melanogaster*) | w[1118]; P{w[+mC]=UAS EcR. B1.dsRNA}168 | Bloomington Stock Center | BDSC:9329 | |
| Genetic reagent (*D. melanogaster*) | w[*]; P{w[+mC]=UAS EcR.A.F645A} TP2 | Bloomington Stock Center | BDSC:9452 | |
| Genetic reagent (*D. melanogaster*) | y1 v1; P{TRiP.JF02257}attP2 | Bloomington Stock Center | BDSC:26717 | |
| Genetic reagent (*D. melanogaster*) | w[1118]; P{w[+mC]=hs-GAL4-EcR.LBD}SBM | Bloomington Stock Center | BDSC:23656 | |
| Genetic reagent (*D. melanogaster*) | w; UAS-GFP-Lsd2 | Michael Welte | | |
| Genetic reagent (*D. melanogaster*) | w;; UAS-EcI RNAi | Naoki Yamanaka | BDSC:37295 | |
| Genetic reagent (*D. melanogaster*) | w1118; P{GD1434}v44851 | VDRC Stock Center | VDRC:44851 | |
| Genetic reagent (*D. melanogaster*) | w1118; P{GD1428}v37058 | VDRC Stock Center | VDRC:37058 | |
| Genetic reagent (*D. melanogaster*) | w1118; P{GD1428}v37059 | VDRC Stock Center | VDRC:37059 | |
| Genetic reagent (*D. melanogaster*) | UAS-E75 RNAi GD | VDRC Stock Center | VDRC:44851 | |
| Genetic reagent (*D. melanogaster*) | UAS-E75 RNAi KK | VDRC Stock Center | VDRC:108399 | |
| Genetic reagent (*D. melanogaster*) | white Canton-S | Laboratory Stocks | | |
| Genetic reagent (*D. melanogaster*) | nSyb-GS[74] | Laboratory Stocks | PMID: 29590612 | |
| Genetic reagent (*D. melanogaster*) | Repo-GS[11] | Laboratory Stocks | | |
| Genetic reagent (*D. melanogaster*) | Actin-GS | Laboratory Stocks | | |
| Genetic reagent (*D. melanogaster*) | w[*]; P{w[+mW.hs]=Switch2} GSG5961 | Laboratory Stocks | | |

*Appendix 1 Continued on next page*

*Appendix 1 Continued*

| Reagent type (species) or resource | Designation | Source or reference | Identifiers | Additional information |
|---|---|---|---|---|
| Genetic reagent (*D. melanogaster*) | w; 9–137 Gal4; TubGal80ts | Laboratory Stocks | | |
| Genetic reagent (*D. melanogaster*) | w; MZ0708-Gal4; TubGal80ts | Laboratory Stocks | | |
| Genetic reagent (*D. melanogaster*) | w; Eaat1-Gal4; TubGal80ts | Laboratory Stocks | | |
| Genetic reagent (*D. melanogaster*) | w; TubGal80ts; NP222-Gal4 | Laboratory Stocks | | |
| Genetic reagent (*D. melanogaster*) | w; NP2222-Gal4 | Laboratory Stocks | | |
| Genetic reagent (*D. melanogaster*) | w; GMR85601-Gal4 | Laboratory Stocks | | |
| Genetic reagent (*D. melanogaster*) | w; Eaat1-Gal4;;, 6 x crossed to Iso | Laboratory Stocks | | |
| Genetic reagent (*D. melanogaster*) | w; GMR56F03-Gal4 | Laboratory Stocks | | |
| Genetic reagent (*D. melanogaster*) | w; NP6293-Gal4, 6 x crossed to Iso | Laboratory Stocks | | |
| Genetic reagent (*D. melanogaster*) | w; Alarm-Gal4 | Laboratory Stocks | | |
| Genetic reagent (*D. melanogaster*) | w; 9–137 Gal4; 6 x crossed to Iso | Laboratory Stocks | | |
| Genetic reagent (*D. melanogaster*) | w; MZ0709-Gal4 | Laboratory Stocks | | |
| Genetic reagent (*D. melanogaster*) | w; moody-Gal4, 6 x crossed to Iso | Laboratory Stocks | | |
| Genetic reagent (*D. melanogaster*) | w; TublinGal80ts/CYO; Repo-Gal4/TM6 | Laboratory Stocks | | |
| Antibody | EcR Antibody DDA2.7 (mouse monoclonal) | Developmental Studies Hybridoma Bank (DSHB) | DDA2.7 (EcR common) | 2 ug/ml |
| Sequence-based reagent | E74_FOR | This paper, Integrated DNA Technologies | PCR primers | TGA GAC GCG AGG AAT ACC CTG GAC |
| Sequence-based reagent | E74_REV | This paper, Integrated DNA Technologies | PCR primers | AAC TGC AGC GT GTA GCC GTT TCC |
| Sequence-based reagent | E75A_FOR | This paper, Integrated DNA Technologies | PCR primers | TCA GCA GGC CAA TCT GCA CCA CTC |
| Sequence-based reagent | E75A_REV | This paper, Integrated DNA Technologies | PCR primers | TGA TGT ACT CGG GAG TCT GGG GAC |
| Sequence-based reagent | E75B_FOR | This paper, Integrated DNA Technologies | PCR primers | AGC AGC ACC AGC ACC AGC AAC AAC |
| Sequence-based reagent | E75B_REV | This paper, Integrated DNA Technologies | PCR primers | ATT GCC CGC ACT GGA GTT GCT CGA |

*Appendix 1 Continued on next page*

*Appendix 1 Continued*

| Reagent type (species) or resource | Designation | Source or reference | Identifiers | Additional information |
|---|---|---|---|---|
| Sequence-based reagent | Ecl_FOR | This paper, Integrated DNA Technologies | PCR primers | TGC AGT GCC GCT CTC AAC TGT ACC |
| Sequence-based reagent | Ecl_REV | This paper, Integrated DNA Technologies | PCR primers | TCA CAG TAA CCG TTG ACC GCC TCC |
| Sequence-based reagent | EcR_C_FOR | This paper, Integrated DNA Technologies | PCR primers | TCA ACC ACA GCC ACA GCT CCT TCC |
| Sequence-based reagent | EcR_C_REV | This paper, Integrated DNA Technologies | PCR primers | TGA TGG GTC CTA TGG CCG CAC TTC |
| Sequence-based reagent | EcR_1_FOR | This paper, Integrated DNA Technologies | PCR primers | GCG GCC AAG ACT TTG TTA AG |
| Sequence-based reagent | EcR_1_REV | This paper, Integrated DNA Technologies | PCR primers | GGC CAA CTG ATT GTA CGT TAA G |
| Sequence-based reagent | EcR_2_FOR | This paper, Integrated DNA Technologies | PCR primers | GCC ATC TGA AGA GGA TCT CAG |
| Sequence-based reagent | EcR_2_REV | This paper, Integrated DNA Technologies | PCR primers | AAC GCT GGT AGA CCT TTA GC |
| Commercial assay or kit | 20-Hydroxyecdysone EIA kit | Cayman | Item No. 501390 | |
| Commercial assay or kit | RNeasy Plus Mini Kit | Qiagen | Item No. 74134 | |
| Chemical compound, drug | BODIPY 493/503 | Fisher | D3922 | 1 ug/ml |
| Chemical compound, drug | 20-Hydroxyecdysone | Sigma | H5142 | 0.5 mM |
| Chemical compound, drug | Gaboxadol | Sigma-Aldrich | T101 | 0.1 mg/ml |
| Software, algorithm | Clocklab | Actimetrics | https://actimetrics.com/ | |
| Software, algorithm | Adobe Illustrator 2020 | Adobe | https://www.adobe.com/ | |
| Software, algorithm | BioRender | BioRender | App.biorender.com | |
| Software, algorithm | Pysolo | *Gilestro and Cirelli, 2009* | https://www.pysolo.net/about/ | |
| Software, algorithm | GraphPad Prism v9 | GraphPad Software | https://www.graphpad.com/ | |
| Software, algorithm | JTK_CYCLE | *Hughes et al., 2010* | Hughes et al., 2010 | |
| Software, algorithm | DAMFileScan113 | Trikinetics | https://trikinetics.com/ | |

