## [Editor Report]

This is a strong manuscript that identified a role for ecdysone signaling and cortex glia in sleep regulation. The manuscript is important because it opens up new avenues of study for examining how hormone signaling and glia regulate sleep and circadian rhythms.

---

## [Decision Letter]

**Decision letter after peer review:**

Thank you for submitting your article "Ecdysone acts through cortex glia to regulate sleep in *Drosophila*" for consideration by *eLife*. Your article has been reviewed by 2 peer reviewers, and the evaluation has been overseen by Mani Ramaswami as Reviewing Editor and K VijayRaghavan as the Senior Editor. The following individual involved in the review of your submission has agreed to reveal their identity: Alex C Keene (Reviewer #1).

Essential revisions:

1) Please provide the necessary details requested regarding background controls and how they may influence conclusions reached. The specific issues pertaining to the RepoGS driver be addressed better by additional experiments using repo-GAL4 combined with GAL80ts, which would provide independent confirmation of the conclusions reached, which would be useful given concerns with the background strains.

2) The effect of Ecdysone perturbations should be shown separately for starved flies and fed flies for reasons explained in Reviewer 2's comment 2.

3) Please make necessary clarifications in the Methods section pertaining to point 2 above.

4) There is. concern that evidence for "mobilization of lipid droplets being the mechanism linking steroid signaling to sleep" is currently quite weak. Additional experiments to strengthen this connection would be valuable, but if these cannot be done, then please qualify the conclusions and modify the discussion to acknowledge the remaining ambiguities in the mechanism by which ecdysone affects sleep.

*Reviewer #1 (Recommendations for the authors):*

I have described a number of concerns, primarily related to interpretations.

1. For the targeted screen in Figure 1, it is surprising that almost all of the genes result in reduced sleep. Is there concern that this may result from background differences? If not, it would seem to deserve additional commentary that all NHRs impact sleep in both neurons and glia.

2. Again, for Figure 1, I very much appreciate the data presentation because it allows all data to be depicted on simple graphs. I do think it would be beneficial to include a Supplemental Figure depicting data with the mean and variance (rather than the differences compared to controls).

3. Line 136: The statement 'perhaps because EcI is one of the most highly expressed genes in glia and may be insufficiently knocked down by RNAi' is very speculative. Suggest directly testing this, or removing the statement. It could also be moved to the discussion.

4. Line 146 states EcR functions specifically in glia to regulate circadian rhythms, but the data also suggest a role in neurons. Suggest changing to 'is required in glia.'

5. Why is the number of arrhythmic flies so high in the repo-GS control?

6. Suggest changing 2I to %, rather than N to allow direct comparisons between groups.

7. I cannot see the blue and green experimental data points in figures 3B and 3C.

8. Is the conclusion from 3E that ecdysone is specifically involved in sleep during starvation, or that glial ecdysone promotes sleep, and the observed sleep loss is cumulative but separate from starvation?

9. Is it possible to change the contract of Figure 5A?

10. Line 282: Since EcR has an effect on neurons, I have some concerns about the interpretations that 'ecdysone affects sleep in LDs …especially in cortex glia'.

---

## [Author Response]

Essential revisions:1) Please provide the necessary details requested regarding background controls and how they may influence conclusions reached. The specific issues pertaining to the RepoGS driver be addressed better by additional experiments using repo-GAL4 combined with GAL80ts, which would provide independent confirmation of the conclusions reached, which would be useful given concerns with the background strains.

We conducted the Repo-Gal4; TubGal80ts>EcR RNAi #1 experiment, as suggested, and the results show that Repo-Gal4; TubGal80ts>EcR RNAi #1 flies sleep a similar amount at permissive temperature (18 degrees) as control flies, but their sleep is significantly less at restricted temperature (31 degrees), confirming that EcR acts in adult glia to regulate daily sleep (Figure 2 —figure supplement 3A, B). Since EcR RNAi #1 and #2 are two independent P-element transgenic lines but were generated using the same construct, we also tested three other EcR RNAi lines with Repo-GS, all of which produced reduced sleep phenotypes, further confirming that EcR RNAi knockdown in glial cells reduces sleep (Figure 2 —figure supplement 3C-E).

We did observe some differences in sleep in GS control flies from experiment to experiment, and this was a concern for us; we agree that Repo-GS flies, and perhaps all GS lines, have higher inter-individual variability. But we do not believe this affects our conclusions for the following reasons. First, we present a lot of data documenting sleep effects of EcR knockdown with Repo-GS. From our initial screening results alone, we had 144 Repo-GS control flies with an average sleep time of 728.9 minutes and an SEM of 13.37. We had 124 *nsyb*-GS control flies with an average of 751.4 min and SEM 14.50. For UAS-EcR RNAi #1 control flies, we had 80 flies with an average of 910.8 min and an SEM of 11.62. Since Repo-GS>EcR RNAi #1 flies had an average sleep time of 460 minutes, despite greater variability in Repo-GS control flies, glial EcR knockdown significantly reduced sleep compared to both control groups. Second, while Repo-GS flies showed greater variance in daytime sleep, their nighttime sleep was more consistent (daytime and nighttime SEM 9.297 vs. 6.404), so we also showed nighttime sleep for our screen, and we chose EcR as a candidate gene based on total sleep and nighttime sleep. Third, while Repo-GS control flies shown in the EcI RNAi experiment slept significantly less than Repo-GS from the initial screen, likely due to experimental variability, the average of 620 minutes was still much larger than the average of Repo-GS>EcR RNAi #1 in the initial screen. Last, data from our new TARGET system and three other EcR RNAis confirm that glial EcR modulates sleep. Overall, despite the small differences in control groups, our experimental groups are always significant compared with the two control groups in each replicate, and we reconfirmed these results by using different systems and different RNAi lines.

2) The effect of Ecdysone perturbations should be shown separately for starved flies and fed flies for reasons explained in Reviewer 2's comment 2.

We have done so. As shown by Hiroshi Ishimoto et al. (2010), we find that 20E promotes sleep in (fed) wild-type flies (Figure 3A, E); and we show that it also does so in starved flies (Figure 3B, E). Thus, ecdysone can promote sleep in both fed and starved flies, and it likely does so independently of starvation.

3) Please make necessary clarifications in the Methods section pertaining to point 2 above.

Methods section has been updated in accordance with reviewers’ requests.

4) There is. concern that evidence for "mobilization of lipid droplets being the mechanism linking steroid signaling to sleep" is currently quite weak. Additional experiments to strengthen this connection would be valuable, but if these cannot be done, then please qualify the conclusions and modify the discussion to acknowledge the remaining ambiguities in the mechanism by which ecdysone affects sleep.

We are happy to report that we conducted additional experiments to strengthen the link between ecdysone and lipid droplets and these support our conclusion. We speculated that since ecdysone feeding mobilizes lipid droplets, glial EcR knockdown should cause flies to accumulate more lipids. Therefore, we performed lipid staining comparing vehicle control and RU486-induced groups of Repo-GS>EcR RNAi flies (new figure 6 —figure supplement 1). We found that induction of EcR knockdown in glial cells with RU resulted in more LDs, whereas RU had no effect on UAS control flies, suggesting that ecdysone signaling bidirectionally affects LDs in glial cells. Increased sleep with ecdysone feeding reduces LDs while reduced sleep with loss of ecdysone signaling increases LDs.

In response to the reviewer’s comment that *lsd2* mutants (which are compromised in LD formation) may not respond to ecdysone because they are incapable of increasing sleep, we fed *lsd-2* mutant flies with gaboxadol, a GABA agonist known to induce deep sleep. Gaboxadol treatment significantly increased the amount of sleep in *lsd-2* mutants (new Figure 6 —figure supplement 2), indicating that *lsd-2* mutant flies can sleep more. Lack of a sleep-promoting effect of ecdysone on *lsd-2* mutants, together with bidirectional effects of ecdysone on LDs, indicate strongly that ecdysone regulates sleep through lipid droplets in glial cells.

Reviewer #1 (Recommendations for the authors):I have described a number of concerns, primarily related to interpretations.1. For the targeted screen in Figure 1, it is surprising that almost all of the genes result in reduced sleep. Is there concern that this may result from background differences? If not, it would seem to deserve additional commentary that all NHRs impact sleep in both neurons and glia.

From Figure 1 and the updated Figure1 – source data 1, it is clear that almost all NHR knockdowns result in varying degrees of sleep reduction. So yes, we agree that most NHRs affect sleep both in neurons and glia. Half of all NHRs respond to ecdysone, so it is not surprising that these ecdysone responsive NHRs interfere with sleep, but why all the other NHRs play a role in sleep deserves further exploration. Since each NHR knockdown line was compared to both its GS control and UAS control, this is unlikely to be a background issue. We discussed this result in the manuscript as suggested.

Surprisingly, we found that knockdown of most NHRs in neurons and glial cells reduced adult sleep to varying levels compared to their GS and UAS controls, suggesting that the endocrine system in general plays a positive role in sleep regulation.”

2. Again, for Figure 1, I very much appreciate the data presentation because it allows all data to be depicted on simple graphs. I do think it would be beneficial to include a Supplemental Figure depicting data with the mean and variance (rather than the differences compared to controls).

We have updated Figure1- source data 1 with mean total/day/night sleep amount and variance for each genotype.

3. Line 136: The statement 'perhaps because EcI is one of the most highly expressed genes in glia and may be insufficiently knocked down by RNAi' is very speculative. Suggest directly testing this, or removing the statement. It could also be moved to the discussion.

We have removed the statement. Based on the transcriptome data from [23], EcI is among the top 50 most enriched surface glia genes and is expressed at 21-fold higher levels in surface glia than in whole brain, but yes, the original statement was speculative. Thanks for the suggestion.

4. Line 146 states EcR functions specifically in glia to regulate circadian rhythms, but the data also suggest a role in neurons. Suggest changing to 'is required in glia.'

Thanks for the suggestion; we have changed the sentence as suggested.

5. Why is the number of arrhythmic flies so high in the repo-GS control?

In the original Figure 2I, about 34% of Repo-GS flies were arrhythmic compared to 21% of *nsyb*-GS flies, and 13% of EcR RNAi #1 flies. We believe the data were affected by one particular repeat of this experiment (as the repeats were pooled in the original figure), and so we compared multiple repeats across genotypes. The new Figure 2I shows that the percentage of rhythmicity is not significantly different between Repo-GS control flies, *nsyb*-GS control flies and EcR RNAi #1 flies. Importantly, despite the variability in the circadian rhythms of Repo-GS flies (as in their sleep behavior), they are still robustly different from experimental (Repo-GS>EcR RNAi) no matter which way the data are analyzed.

6. Suggest changing 2I to %, rather than N to allow direct comparisons between groups.

Figure2I was updated and please see explanation above.

7. I cannot see the blue and green experimental data points in figures 3B and 3C.

Sorry for the confusion. For Figure 3C and 3D, there are only two groups, fed and starved, of *nSyb*-/Repo-GS>EcR RNAi #1 flies; in the original version the black curve was the fed group, and the red curve was the starved group. In the updated manuscript, we changed the colors in Figure 3C and 3D, hopefully this time it is clear.

8. Is the conclusion from 3E that ecdysone is specifically involved in sleep during starvation, or that glial ecdysone promotes sleep, and the observed sleep loss is cumulative but separate from starvation?

We believe the effect of ecdysone on sleep is independent of starvation, but the role of glia in effects of ecdysone is especially evident during starvation (Figure 3E).

9. Is it possible to change the contract of Figure 5A?

Yes, we changed the light grey color in Figure 5 to a darker grey.

10. Line 282: Since EcR has an effect on neurons, I have some concerns about the interpretations that 'ecdysone affects sleep in LDs …especially in cortex glia'.

That’s a good point, and we have rephrased the sentence as “Ecdysone affects sleep in large part by modulating LDs accumulated in glial cells, especially cortex glia.’